# Early control of viral load by favipiravir promotes survival to Ebola virus challenge and prevents cytokine storm in non-human primates

Stéphanie Reynard[1,2], Emilie Gloaguen[3], Nicolas Baillet[1,2], Vincent Madelain[3], Jérémie Guedj[3], Hervé Raoul[4], Xavier de Lamballerie[5], Jimmy Mullaert[3], Sylvain Baize[1,2]*

**1** Unité de Biologie des Infections Virales Emergentes, Institut Pasteur, Lyon, France, **2** CIRI, Centre International de Recherche en Infectiologie, Univ Lyon, Inserm, U1111, Université Claude Bernard Lyon 1, CNRS, UMR5308, ENS de Lyon, Lyon, France, **3** Université de Paris, IAME, INSERM, Paris, France, **4** Laboratoire P4 Jean Mérieux–INSERM, INSERM US003, Lyon, France, **5** Unité des Virus Émergents (UVE Aix-Marseille Université-IRD 190-Inserm 1207-IHU Méditerranée Infection), Marseille, France

* sylvain.baize@pasteur.fr

**Data Availability Statement:** Data from proteomic studies ara available through Zenodo, DOI 10.5281/zenodo.4611597.

## Abstract

Ebola virus has been responsible for two major epidemics over the last several years and there has been a strong effort to find potential treatments that can improve the disease outcome. Antiviral favipiravir was thus tested on non-human primates infected with Ebola virus. Half of the treated animals survived the Ebola virus challenge, whereas the infection was fully lethal for the untreated ones. Moreover, the treated animals that did not survive died later than the controls. We evaluated the hematological, virological, biochemical, and immunological parameters of the animals and performed proteomic analysis at various timepoints of the disease. The viral load strongly correlated with dysregulation of the biological functions involved in pathogenesis, notably the inflammatory response, hemostatic functions, and response to stress. Thus, the management of viral replication in Ebola virus disease is of crucial importance in preventing the immunopathogenic disorders and septic-like shock syndrome generally observed in Ebola virus-infected patients.

## Author summary

Ebola virus was responsible for several epidemics in the recent years and is now considered as a major public health concern in Central and West African countries. We and others demonstrated that pathogenic events observed during Ebola virus disease are linked to a deleterious immune response. However, the mechanisms implicated are not fully understood. Here, we studied immune responses depending on the viral loads observed in infected cynomolgus monkeys. An antiviral treatment allowed the reduction of viral load in some animals and we observed that these animals did not experience deleterious immune response and the loss of hemostasis. The release of pathogen-associated

**Funding:** The authors received no specific funding for this work.

**Competing interests:** The authors have declared that no competing interests exist.

molecular patterns may thus be limited by the inhibition of viral replication, avoiding the overstimulation of the immune system and consequently the pathogenic events observed in Ebola virus disease.

## Introduction

West Africa suffered from the biggest Ebola virus (EBOV) epidemic to date between 2014 and 2016, resulting in 28,652 cases and 11,325 deaths. This led the WHO to launch a fast-track process to identify potential drug treatments against EBOV. In this context, favipiravir was tested for its benefit against EBOV infection. A clinical trial in the field showed only limited efficacy of the treatment. However, the measured plasma concentrations of favipiravir were lower than expected, which could have affected treatment efficacy [1,2].

The efficacy of favipiravir against EBOV infection has also been evaluated in non-human primates (NHPs), the gold standard model for in vivo pathogenesis studies [3]. Although some of the experiments did not show a benefit for survival, the time to death was at least delayed and associated with a lower viral load than that in non-treated monkeys [4–6]. Significant survival was obtained with intra-venous treatment only. However, the reason why the reduction of virus replication did not allow a better improvement of survival was not fully elucidated.

The pathogenesis of EBOV has been associated with defective immune responses and the release of numerous cytokines [7–11]. A deleterious role for the immune response has been proposed, supported by the poor outcome, which is associated with the non-regulated synthesis of pro-inflammatory and anti-inflammatory cytokines, chemokines, and cytotoxic compounds [8,11,12]. By contrast, survivors have shown well-balanced immune responses, with transient expression of these molecules. The dysregulated synthesis of cytokines in fatal cases was defined as a "cytokine storm" and may be involved in impairment of the endothelium, leading to the vascular leakage and intravascular coagulation observed in Ebola virus disease (EVD) [13,14].

We studied the potential correlations between the cytokines expressed, outcome, and viral load to decipher the mechanism by which favipiravir and immune responses affect the disease outcome. We used samples from a previous experiment in which three doses of favipiravir were tested on NHPs infected with EBOV, the two highest doses resulting in 40 and 60% survival [5]. Biochemical, virological, and immunological parameters were measured in the favipiravir treated- and untreated-infected NHPs to establish the potential impact of reducing the viral load on the immune response.

## Methods

### Ethics statement

Animal experiments received the agreement of the ethical committee of Rhone-Alpes, registered with the French Ministry of Research (number 2017APAFIS#6097–2016062713281115 and CECCAPP C2EA15) and were performed in accordance with the European guidelines for animal facilities.

### Non-human primate experiment and sampling

Briefly, plasma was taken from 20 cynomolgus monkeys infected with 1,000 FFU of EBOV (Ebola Gabon 2001 strain) and treated, or not, with favipiravir, either 150 mg/kg or 180 mg/kg twice a day for 14 days and starting two days before infection [5,6]. All non-treated animals

(n = 10) were euthanized between days 7 and 10, as they had reached the pre-determined clinical endpoint (5). Five treated animals were euthanized between days 10 and 19 (three animals treated with 150 mg/kg and two with 180 mg/kg). The five remaining animals survived until the pre-determined endpoint of the experiment, at day 21 post-infection (two animals treated with 150 mg/kg and three animals treated with 180 mg/kg). One of the treated animals was euthanized during the course of the protocol because of a wound on the leg that presented signs of necrosis with no evident link with the treatment or the EBOV infection. It was thus removed from the study to avoid inducing a bias in the immune response observed, as the endpoint was not reached because of EBOV infection.

For flow cytometry experiments, whole blood from 10 of the animals was used (five untreated and five treated animals, including three survivors) and five animals from another experiment (including one survivor) treated with 180 mg/kg twice a day from day 0 to day 14.

The data of viral load, IFNα, IL6 and TNFα presented in this article have been previously reported in Madelain et al, 2018 [6], but this study aims at studying the outcome of the disease, we thus needed to present once more the data depending on our criteria of analysis.

## Viral loads and biochemical and hematological parameters

Viral loads and viral titers were measured as previously described [5,6]. Hematological and biochemical analyses were performed at each sampling point using a Pentra C200 analyzer (Horiba) and a MS9-5s Hematology analyzer (Melet Schloesing Laboratories).

## IFNα ELISA

Cynomolgus IFN-α2 was detected by ELISA in the BSL4 laboratory using matched paired antibodies and a protein standard (Life Technologies), according to the manufacturer's instructions.

## Cytokine/chemokine multiplex assays

In the BSL4 laboratory, 37 parameters were assayed in NHP plasma samples using NHP cytokine magnetic-bead panels I and II (Merck). Plates were prepared according to the manufacturer's recommendations and read on a Magpix instrument (Merck).

## Flow cytometry

Fresh whole blood was stained with surface antibodies. At day 0, cells were stained with CD3, CD4, CD8, and CD69 (BD Biosciences) for 30 min on ice before red cell lysis and fixation using Immunoprep reagents (Beckamn Coulter). At day 3, we observed that the favipiravir molecule was fluorescent and emitted in the channels associated with the 405 nm laser. Thus, we were unable to analyze these samples and changed the panel of antibodies for the following points to avoid detecting the fluorescence of the treatment. From day 5, CD3, CD4, and CD69 were used (only the CD3 fluorochrome changed and was used in place of CD8). Cells were analyzed using a Gallios analyzer (Beckman Coulter).

## Proteomic analysis

Plasma samples from days 0, 3, and 10 (or 9 when day 10 was not available) were used. They were prepared and analyzed as previously described [15]. Briefly, plasma samples were depleted of albumin and immunoglobulins and then analyzed using nano LC-MS/MS technology.

## Statistical analysis

Correlations between viral load and protein expression were calculated with Sigma plot software using a Pearson product moment correlation.

All other statistical analyses were performed using R 3.6.1 software [16] and graphs using the *ggplot2* R-package [17].

For the Luminex experiments, protein data were analyzed only if more than 20% of the NHPs had results available. A first comparison was performed between the mean protein levels on day 0 and day 7 for all NHPs using the Wilcoxon rank-sum test. Then, proteins for which the level differed ($p \leq 0.10$) were compared between three groups (control, fatal, nonfatal) using the Kruskal–Wallis test ($p \leq 0.05$). Principal component analysis (PCA) was performed to summarize the information contained in the data for the retained proteins by reducing its dimensionality using the *factoMineR* R-package [18].

For proteomic data, proteins with less than two peptides, including one specific peptide, and those detected for less than four NHPs at each timepoint were excluded. Spectral counts of each protein were log2 transformed and normalized using the quantile normalization method of the *limma* R-package [19].

The viremia level (VL) on day 7 was log10 transformed and dichotomized into two groups (high VL for $Log10(VL) > 5$ and low VL for $Log10(VL) \leq 5$). Normalized spectral counts were analyzed using linear mixed effect models with the *lme4* R-package [20]. Two models were compared using a likelihood ratio test (LRT), one with a random effect of the NHP and only a fixed effect of time and the second adding an interaction with the VL group and time. The effect of each parameter was analyzed to determine which part of the regression was significant. Due to the inflation of the p-value of this test, the empirical p-value was calculated by performing 100,000 permutations of the analysis using the same strategy, each permutation corresponding to reallocation of the VL group for each NHP, without changing the protein data, for each timepoint.

Various biological pathways were defined and the proportions of proteins in these pathways for the entire protein panel or that among proteins selected by the LRT tests (empirical p-value without BH correction $< 0.05$) were calculated. The two proportions were compared using Fisher's exact test.

## Results

### Viremia and hematological and biochemical parameters

To investigate the efficiency of favipiravir antiviral treatment on EBOV infection, the viral loads were measured in plasma samples. Hematological and biochemical parameters were also of interest. Hepatic and renal markers are highly upregulated during EBOV infections together with deep lymphopenia and thrombocytopenia. The Cynomolgus monkeys were thus classified depending on their status (untreated control or favipiravir treated) and outcome (euthanized during the course of the experiment because they reached the clinical endpoint or surviving the disease). Viral loads and viral titers are presented for each group. Viremia reached its peak at day 7 for controls and at day 10 for the treated animals (Fig 1A). However, the treated animals that survived presented more heterogenous viral loads, including an animal that never showed detectable infectious particles despite viral RNA was found in all plasma samples. Viremia was lower with favipiravir treatment, as previously described [5,6].

Profound lymphopenia and thrombocytopenia were observed in each group (Fig 1B). In fatally-infected animals, lymphocyte counts dropped until euthanasia, except for a few animals for which they started to rise just before reaching the clinical endpoint. Lymphopenia was also

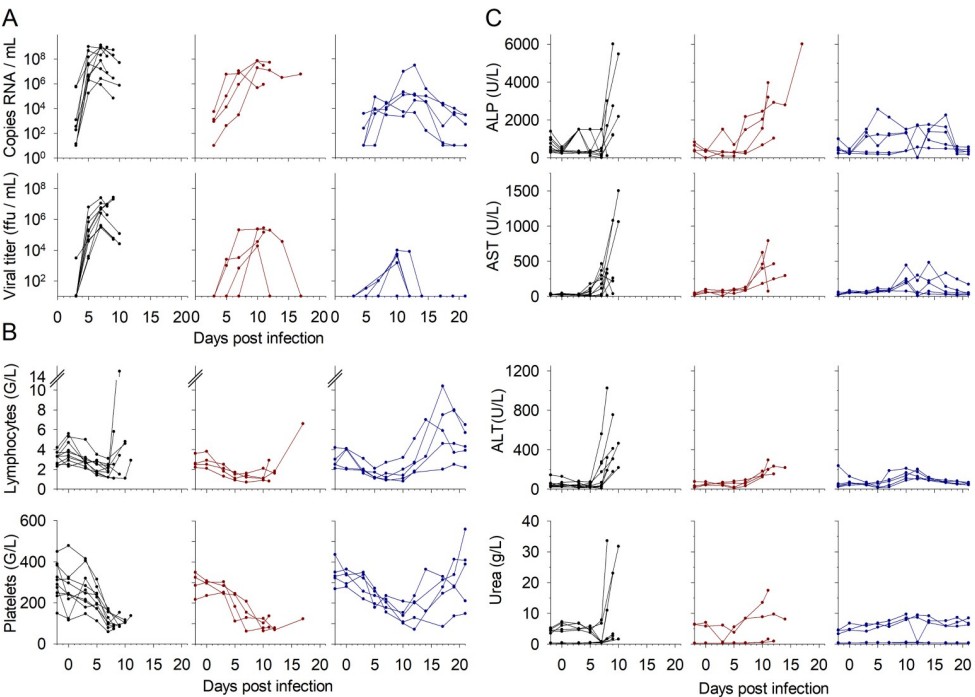

**Fig 1. Virological, hematological, and biochemical parameters.** Individual values from control (black, n = 10), treated and fatally-infected (red, n = 4), and treated and surviving (blue, n = 5) animals were measured during the course of the experiment and are presented. A. Viral loads and viral titers were determined by RT-PCR and plaque assay, respectively, to determine the absolute values of genome copies and infectious particles. B. Lymphocyte and platelet counts were determined using a hematological analyzer and expressed as Giga/L ($10^9$/L). C. The prototypic hepatic enzymes ALP, AST, and ALT were determined, along with blood urea nitrogen, to evaluate liver and kidney function during EVD.

observed until day 10 in the survivors, but clear lymphocytosis began from day 12 and was not fully resolved by day 21. Thrombocytopenia perfectly matched the viremia, the lowest platelet counts being observed at days 7 and 10 for untreated and treated animals, respectively. Thrombocytosis also appeared in survivors from day 12.

Biochemical analyses were performed throughout the experiment. Hepatic enzymes were strongly expressed on the last days before euthanasia in the animals that died from the infection (Fig 1C). Several of the control animals showed extremely high levels of these hepatic markers in the late stage of the disease, whereas only one of the treated and fatally-infected animals expressed very high levels of ALP only. On the contrary, survivors did not show high levels of these enzymes.

In addition, several control animals also expressed high levels of plasma urea, indicating dehydration, kidney injury or hemorrhage.

## Inflammatory response

Inflammatory markers were measured using a multiplex technology, except for IFNα, which was tested by ELISA. All control animals strongly expressed IFNα and TNFα (Fig 2A). The other pro-inflammatory cytokines tested were expressed more heterogeneously but increased markedly in several animals before euthanasia. The treated animals with a lethal infection also synthesized pro-inflammatory cytokines, but at significantly lower levels and later (from day 7 vs day 5 for controls). The animals that survived expressed pro-inflammatory cytokines at moderate levels, except for IL6 and IL18, which were not detectable in most of the samples

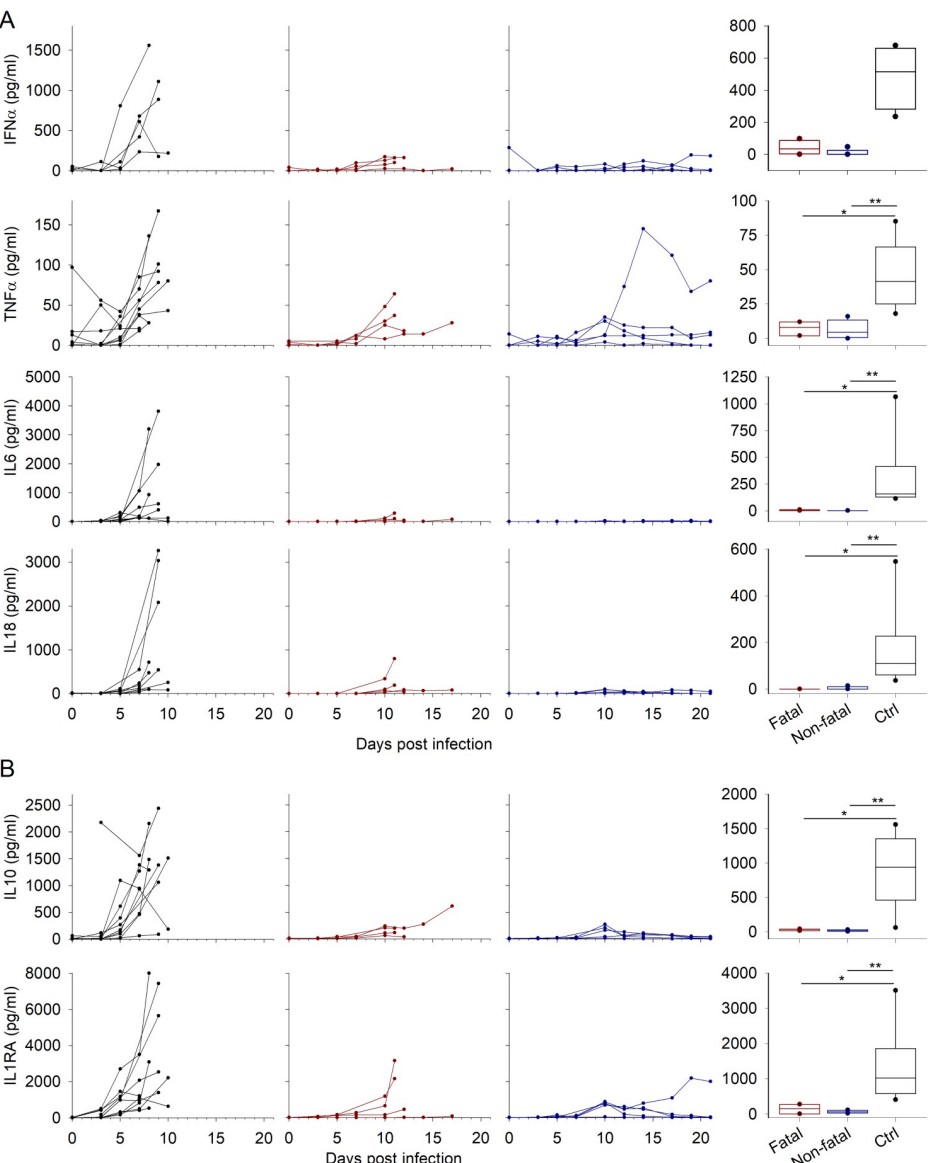

**Fig 2. Inflammatory and anti-inflammatory mediators.** A. Inflammatory soluble mediators were measured at each sampling point during the course of the experiment by Luminex assay or ELISA for IFNα. The animals tested and the representation of the values are the same as in Fig 1. The whisker plots present the dispersion of values at day 7. P-values are indicated for pairwise comparisons (*p < 0.05; **p < 0.01). Ctrl corresponds to the control NHPs, fatal and non-fatal correspond to treated NHPs, depending on the outcome. B. Anti-inflammatory soluble mediators were analyzed and are presented as in A.

tested or at very low levels. Interestingly, one animal had high levels of TNFα, which peaked at day 14 and remained high until the end of the protocol (day 21).

The anti-inflammatory cytokines IL10 and IL1-RA were also strongly expressed in control animals (Fig 2B). High amounts were synthesized from day 7, conjointly with the pro-inflammatory molecules. Treated animals secreted significantly lower levels of these molecules. We observed only slight expression on day 10 in survivors.

Chemokine expression also depended on the treatment status. The control animals synthesized significantly higher levels of chemokines on day 7 than those that were treated, except for

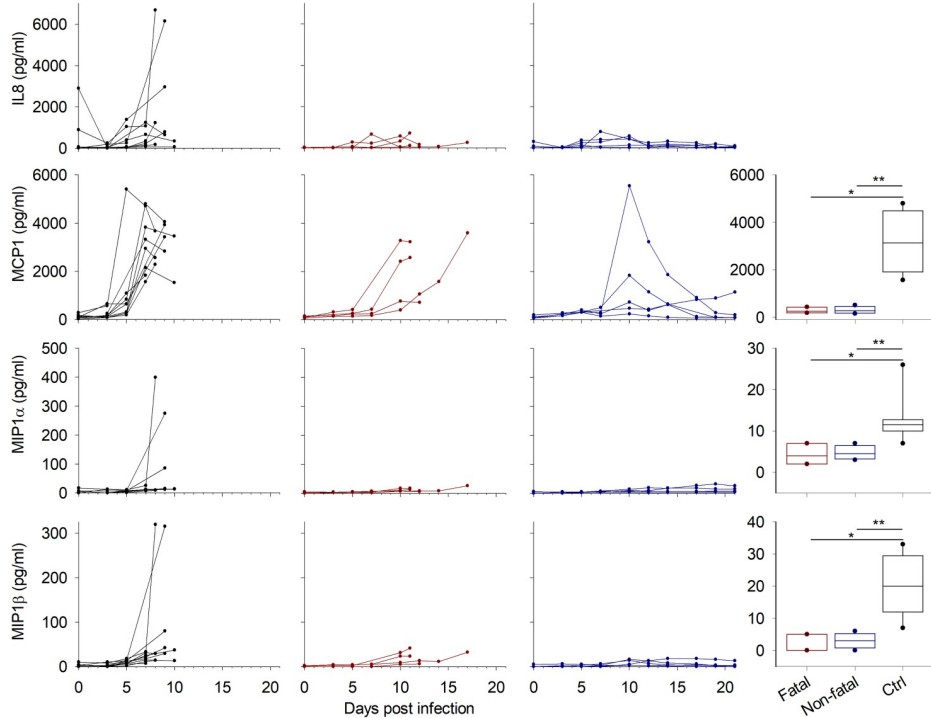

**Fig 3. Chemokine expression.** Chemokines were measured by Luminex assay and the results are presented as in Fig 2.

IL8, which was not strongly expressed in most of the controls. Only three of ten animals showed high amounts of this protein, but only one sample was available on day 7 and thus included in the group comparison (Fig 3). MCP1 was the only chemokine tested that was massively produced in all control animals. Plasma levels of this molecule were also elevated in the treated and fatally-infected animals from day 10, instead of day 7 in controls. Those who survived showed only a peak at this timepoint. The other chemokines tested were not strongly expressed in the treated animals.

## T-cell response

Cytokines responsible for the activation, differentiation, and proliferation of lymphocytes were significantly over-expressed in the final days before death in the controls, peaking at day 7 (Fig 4A). CD69, a marker of T-cell activation, non-specific for the antigen, was also transiently expressed at this timepoint. Lower expression of T-cell cytokines and CD69 was also observed in treated animals. A delayed cytokine peak appeared on day 10 in these animals, whereas it remained on day 7 for CD69. All cytokines tested returned to basal levels in the survivors, except for one animal that maintained IL2 expression until the end of the experiment.

Cytotoxic molecules were also differentially expressed between control and treated animals (Fig 4B). IFNγ and GrzB remained at very low levels in the favipiravir-treated animals throughout the course of the experiment, whereas they were over-expressed in most of the control animals. The expression of GrzB and sFasL began from day 7 in the control animals, making it impossible to perform a significance test at day 7, the peak of viremia. However, sFasL also tended to be more highly released in the control animals. By contrast, perforin was strongly expressed by both groups but the timing of expression differed in the survivors (from day 5 vs day 3). The animals who died from the infection expressed increasing levels of

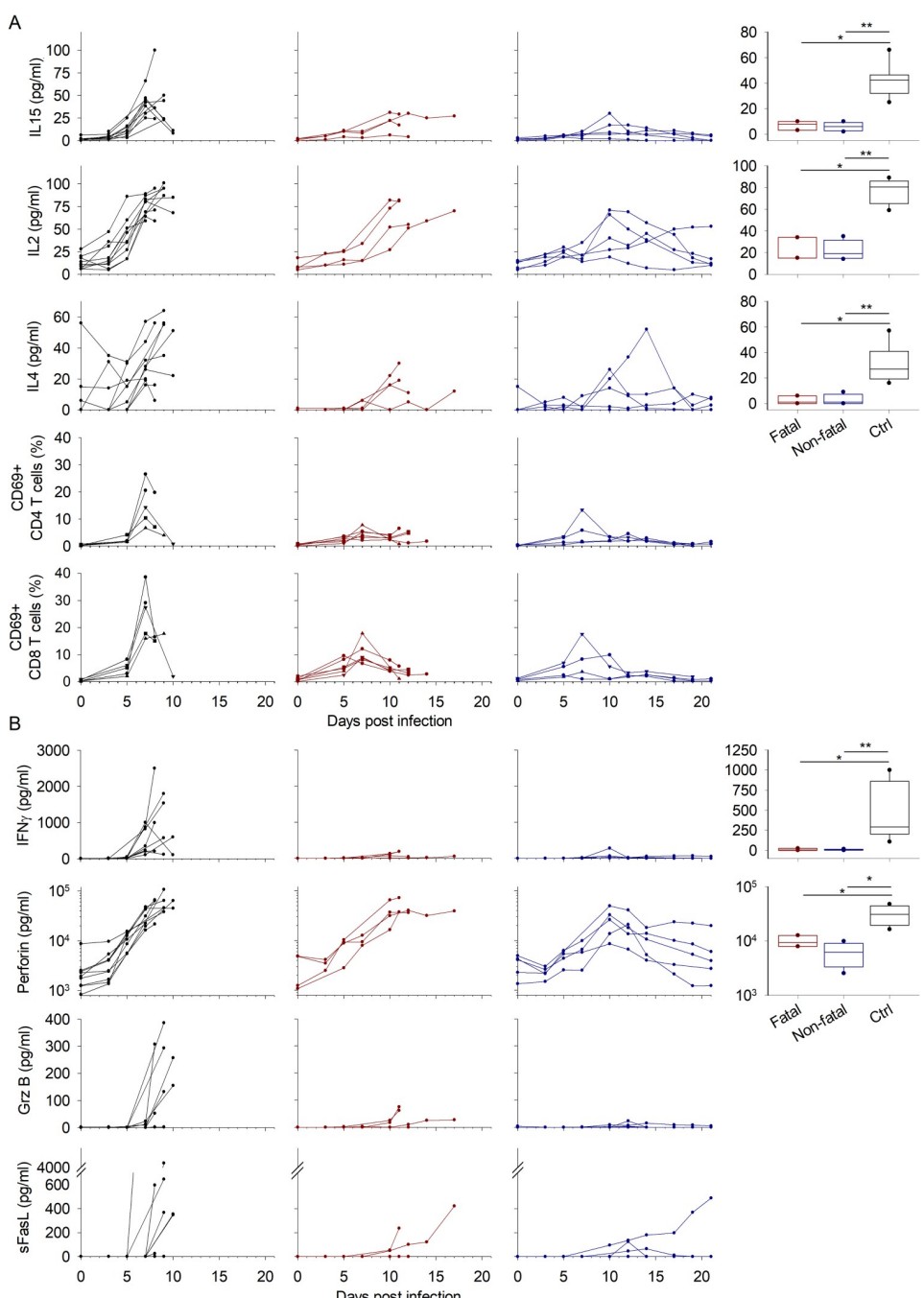

**Fig 4. T-cell response.** A. Cytokines involved in the T-cell response are presented as in Fig 2. The activation of T lymphocytes was evaluated by flow cytometry. CD69 expression was calculated for CD4 and CD8 T-cell populations. Results are presented as for the Luminex assay. B. Soluble cytotoxic molecules were assayed and are presented as in Fig 2.

perforin until death. Treated animals showed significantly lower expression at day 7 than control animals. Perforin values for most of the surviving animals returned to normal after the peak at day 10. The expression of the co-stimulatory molecule sCD40L was also evaluated (S1 Fig) and this molecule tends to increase in the days before death in control animals.

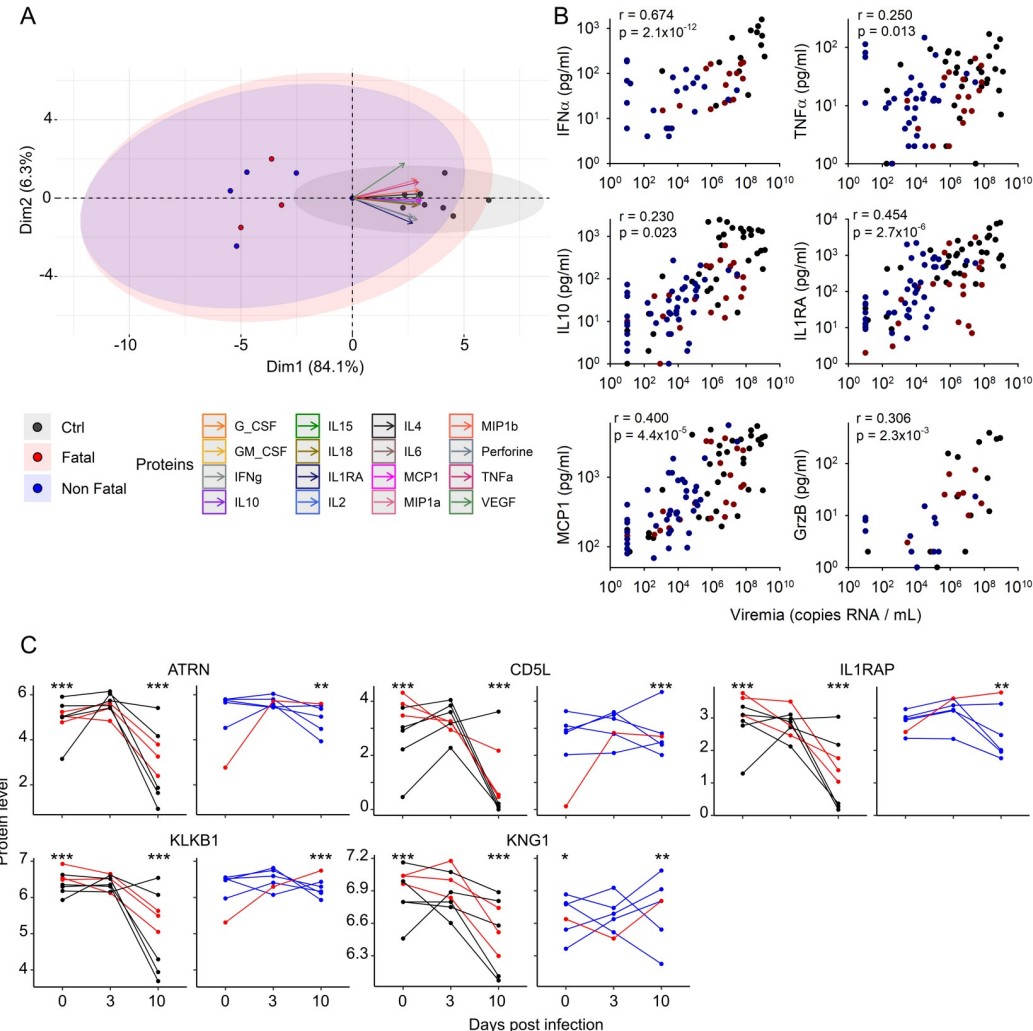

**Fig 5. Correlation between viremia and cytokine expression.** A. PCA, including results from all parameters analyzed and for which measurable values were obtained, is presented. Control animals (black), treated animals with a fatal infection (red), and treated animals who survived (blue) are presented individually. The trend for the evolution of the protein level is presented by an arrow. B. Dot plots presenting the correlation between the viral load and protein level. Protein values from each animal are plotted for each timepoint. Control animals are indicated by black dots, fatally-infected and treated animals by red dots, and survivors by blue dots. A Pearson correlation coefficient and the associated p value was calculated and is presented in each plot. C. Proteomic analysis was performed on plasma samples from days 0, 3, and 10. Proteins involved in the inflammatory pathway and significantly regulated between group and timepoint are represented (ATRN: Attractin, CD5L: CD5 molecule like, IL1RAP: IL1 receptor accessory protein, KLKB1: Kallikrein B1 and KNG1: Kininogen 1). The evolution of the protein level was plotted depending on the level of viremia (left: high viremia, right: low viremia). The colors of the dots and lines are as in B. P values were calculated depending on timepoint and group (low or high viremia) and are indicated (*p < 0.05; **p < 0.01; ***p < 0.001). The p value stated at day 0 of the high-viremia group (left) indicates that the protein level actually evolves depending on the viremia and/or the time point. For the graph of low viremia group (right), the p value at day 0 indicates the difference on day 0 between the low- and high-viremia groups. For days 3 and 10, the p values indicate the significance of the change relative to day 0 inside the group.

## Cytokine expression is correlated to viremia

All proteins tested that were detectable in the plasma samples were included in a PCA (Fig 5A). Values obtained at day 7 were used, corresponding to the peak of viremia in control animals. Certain samples were missing for this timepoint and the corresponding subjects excluded to avoid inducing a bias: viremia and protein levels evolve very quickly in this phase

of the disease. Control animals appeared to be clearly clustered, with the treated animals separated from them but appearing to be more heterogeneous. The cytokine storm was consistently associated with uncontrolled viral replication. Moreover, the peak of expression of these cytokines was shifted from day 7 in the controls to day 10 in the treated animals, as were viremia levels. Thus, we plotted the concentrations of the prototypic inflammatory and anti-inflammatory cytokines, chemokine, and cytotoxic molecule IFNα/TNFα, IL10/IL1RA, MCP1, and GrzB, respectively, in the plasma as a function of the viral load for each animal and each time-point (Fig 5B). The expression of all these molecules significantly and positively correlated with viremia. Cytokine levels are therefore correlated with viral replication. We performed a proteomic analysis of plasma samples by liquid chromatography-mass spectrometry to further study the biological pathways involved in the physiopathology associated with EVD. Three time points were studied: day 0 as a baseline for each subject, day 3 for early biological events, and day 10 for the acute phase in fatally-infected individuals. The direct correlation between viral load and protein level led us to analyze the proteomic results as a function of viremia. Animals were therefore divided into two groups: high (Log10 > 5) and low (Log10 ≤ 5) viremia, based on the value at day 7. All control animals were obviously included in the high-viremia group, along with three of four treated and fatally-infected animals. The five animals that survived and the treated animal that survived the longest (until day 19), comprised the low-viremia group. Six of 17 proteins in the inflammatory pathway were significantly differentially expressed depending on the group. A heatmap representing the protein expression highlights the heterogeneity among subjects, along with differences in the trends in the evolution of protein expression (S2 Fig). The proteins that were differentially expressed are represented individually for each timepoint (Fig 5C). These proteins are involved in the regulation of chemotactic activity (attractin) and the mechanisms of inflammation (CD5-like molecule), the inhibition of the pro-inflammatory effect of the cytokine IL1 (IL1R accessory protein), and coagulation (kallikrein and kininogen). The values at day 0 did not differ between the two groups, except for KNG1, which was slightly under-expressed in the low-viremia group. The regulation of these proteins was most significant in the high-viremia group. Indeed, the evolution of the protein level, while significant, did not show a clear tendency in the low-viremia group, but a highly significant drop was observed on day 10 in the plasma of the high-viremia animals.

## Pathogenic effects and tissue damage depend on the viral load

Proteomic analysis of plasma revealed the modulation of pathways that could explain certain pathological events in EVD. First, the coagulation pathway tended to be differentially regulated, depending on the level of viremia. The heatmap shows the expression of the coagulation factors to be quite homogeneous between animals, which tended to be globally down-regulated when viremia was high (S3 Fig). Ten of 29 proteins were significantly differentially expressed in this pathway. Values on day 0 did not differ between the two groups, except for KNG1 and the fibrinogen gamma chain (FGG), which were slightly under and over-expressed, respectively, in the low-viremia group (Fig 6A). On day 3, the evolution of expression was highly heterogeneous for certain proteins, perhaps because of differences in the time of response depending on the individual. However, the levels of most of these proteins were significantly and dramatically altered in high-viremia subjects in the late phase of the disease. The level dropped for eight of ten of the proteins. The two remaining proteins, coagulation factor V (F5) and FGG (fibrinogen gamma chain) where instead upregulated. The levels of coagulation factors were also significantly modified, but more moderately, in low-viremia samples.

In addition, the complement pathway appeared to be highly altered during the disease. Proteins involved in this pathway were globally differentially regulated depending on the

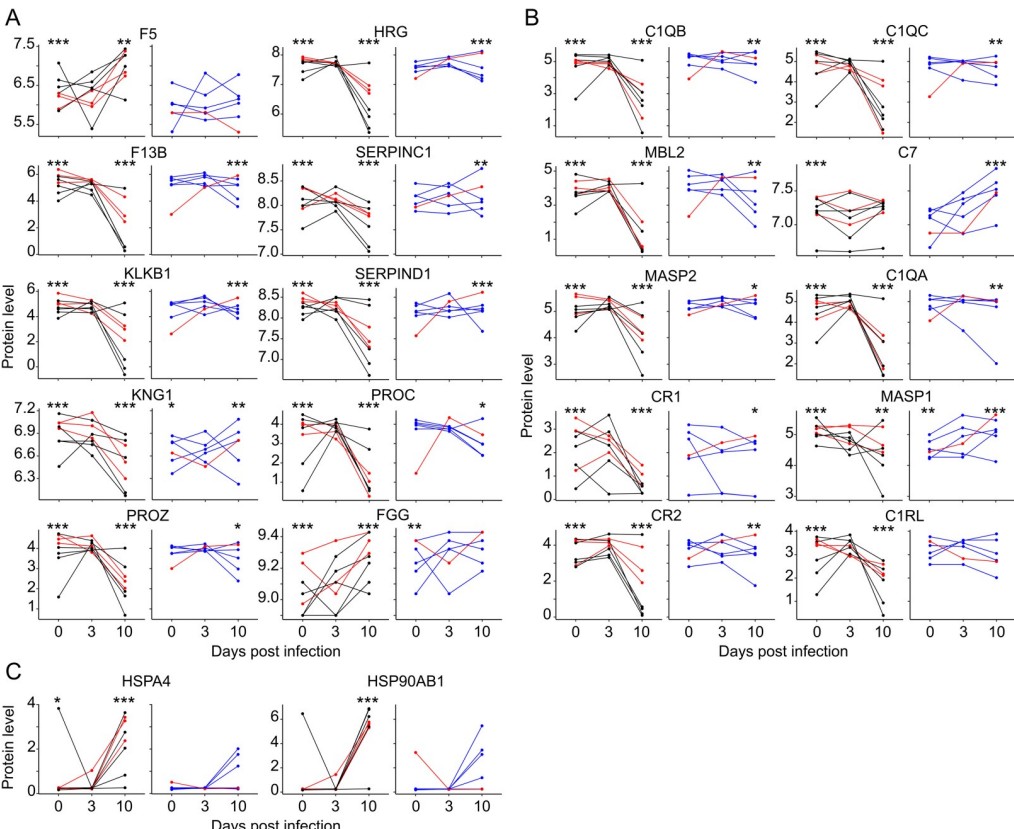

**Fig 6. Dysregulation of pathways involved in homeostasis.** A. The coagulation pathway was studied based on the results of the proteomic analysis. The proteins significantly regulated in this pathway are represented as in Fig 5C (F5: Coagulation factor V, HRG: Histidine Rich Glycoprotein, F13B: Coagulation factor XIII B chain, SERPINC1: Serpin family C member 1, KLKB1: Kallikrein B1, SERPIND1: Serpin family D member 1, KNG1: Kininogen 1, PROC: Anticoagulant protein C, PROZ: Protein Z and FGG: Fibrinogen gamma chain). B. The complement pathway was also analyzed and the results are presented as in Fig 5C (C1QB: Complement C1q B chain, C1QC: Complement C1q C chain, MBL2: Mannose Binding Lectin 2, C7: Complement C7, MASP2: MBL-assocoated serine protease 2, C1QA: Complement C1q A chain, CR1: Complement Receptor Type 1, MASP1: MBL-associated serine protease 1, CR2: Complement Receptor type 2 and C1RL: Complement C1r subcomponent like). C. Heat-shock proteins that were significantly regulated in the stress-response pathway are represented as in Fig 5C (HSPA4: Heat Shock 70kDa Protein 4 and HSP90AB1: Heat Shock Protein 90kDa Protein 1 beta).

level of viremia (p = 9.6x10$^{-4}$ by the permutations calculation). Expression of the complement factors was dramatically downregulated in the animals with uncontrolled viremia (S4A Fig). The levels of proteins that were significantly differentially expressed did not differ on day 0 between the two groups, except for MASP1 (mannan binding lectin serine peptidase 1), which was slightly under-expressed in the low-viremia group (Fig 6B). The levels of all but one protein showed a highly significant drop on day 10 in the subjects with a high viral load. Their expression in the low-viremia group, while significantly modified, did not evolve with a distinguishable trend. The exception was complement component 7 (C7), which tended to be regulated only in this group, with overexpression at day 10, but not in the high-viremia subjects.

Finally, we examined the stress response pathway. Indeed, the proteins involved in this pathway tended to be upregulated, more intensively in highly viremic animals than those in the other group (S4B Fig). Two heat-shock proteins were significantly differentially expressed by the permutations calculation and were significantly upregulated during the late phase of the

disease in high-viremia animals, whereas this change was less pronounced and non-significant in the other animals (Fig 6C).

## Discussion

Here, we studied the host response during EBOV infection in NHPs according to favipiravir treatment and outcome to understand how such treatment modulates the pathogenesis of EVD. Previous studies showed that patients who succumb to EVD present dramatic and relentlessly increasing viral loads, together with an exacerbated release of soluble immune mediators. By contrast, survivors benefit from lower viral replication and a balanced immune response, illustrated by early and well-regulated expression of cytokines and chemokines [7,8,10–12,21–23].

We confirm that fatal EVD is associated with the massive release of inflammatory and anti-inflammatory cytokines and chemokines, reminiscent of septic-shock syndrome. This dysregulated inflammatory response is directly responsible for some if not most of the pathological events that characterize fatal EVD, such as vascular leakage and multiorgan failure [13,24]. We therefore observed high variability on some cytokine expressions in control animals, IL8 for example. This could be due to the endpoint of the experiment: some of the animals could have been euthanized because of their clinical status but just before the intense release of cytokines that precedes death. We also show that severe EVD is accompanied by the intense release of T-cell and cytotoxic cell-related cytokines and soluble factors. This observation indicates that a cellular immune response is induced but fails to control viral replication and suggests that immunopathogenesis linked to T cells and cytotoxic cells can contribute to EVD. Similar results have been reported during severe EVD in humans [11,25,26].

Favipiravir treatment of NHPs leads to a significant decrease in the viral load, even in animals that succumbed to the disease. In these latter animals, the time of survival was nevertheless extended according to the drop in viral load [6]. In our study, the levels of soluble mediators related to inflammatory and T-cell responses correlated with viral loads. These results demonstrate that reduction of the viral load by favipiravir treatment correlated with decreased cytokine/chemokine release. Half of the treated animals survived acute EVD, whereas all untreated animals succumbed. Acting on viral replication and the resulting cytokine/chemokine storm is thus crucial for decreasing the severity of EVD. As previously described [11], a probable hypothesis to explain the pathogenic cascade during EVD is that the absence of the early control of viral replication leads to systemic spreading and an explosive increase in viral load. Due to the pantropism of EBOV, massive viral replication occurs, resulting in the considerable release of pathogen-associated molecular patterns (PAMPs), such as viral RNA and EBOV glycoprotein. The released PAMPs then induce intense activation of inflammatory cells, with the release of soluble mediators, which in turn provoke most of the damage and organ failure characteristic of EVD. Favipiravir is an inhibitor of viral RNA polymerase [27], we thus hypothesize that it decreases the viral load and limits the flood of PAMPs and as a consequence, the intensity of the host response to PAMPs, resulting in lower inflammatory responses and reduced pathology as observed in patients that naturally present lower symptoms and recover [28]. The lower levels of markers related to hepatic and renal failure detected in favipiravir-treated NHPs, even those who died, than in untreated animals are consistent with this hypothesis. Moreover, proteomic analysis highlighted the downregulation of proteins involved in the regulation of the inflammatory response (attractin, IL1RAP, CD5L) in the animals with the highest viremia. Overall, these data show that the cytokine/chemokine storm correlates with the excessive viral load. However, the improvement of clinical and inflammatory parameters was not sufficient to allow all animals to recover from acute EVD. In

contrast to the levels of inflammatory response-related mediators, which uniformly decreased after favipiravir treatment, T-cell and cytotoxic response-related markers were only partially affected in the treated animals. Indeed, whereas activation of CD4+ T cells and the levels of IL-15, IFNγ, GrzB, and sFasL substantially dropped, the levels of IL-2, IL-4, and perforin and the percentage of activated CD8+ T cells were only partially diminished in favipiravir-treated animals. If this cellular response plays a role in the pathogenesis of EVD, its persistence after treatment may partially explain the incomplete efficacy of favipiravir. Interestingly, the expression of soluble CD40L observed in humans in the late phase of the disease has been associated with recovery [11,14]. It is implicated in CD4 T cell response but was not evident in our study (S1 Fig). A very slight expression was observed in the acute phase of the disease in control animals. This discrepancy between humans and NHP could be of interest to investigate.

Although the pathogenic disorders were tightly linked to the inflammatory response, the additive effect of dysregulation of the coagulation and complement pathways and the response to stress must be considered. A close relationship between inflammation and coagulation has been reported and may provide the environment responsible for the disseminated intravascular coagulation observed in EVD [29,30]. In particular, we observed the downregulation of HRG, a protein that can interact with numerous coagulation factors. This protein is degraded in the context of intravascular immunothrombi [31] and its loss has previously been described to be a strong marker of sepsis [32]. Not only is the complement pathway involved in the pathogenesis of EBOV through antibody-dependent enhancement (ADE) [33,34], it is also involved in the severity of other viral infections and inhibition of the T-cell response [35]. Indeed, the complement and coagulation pathways can activate each other and are closely related [36]. The loss of proteins from the complement pathway in plasma samples may be a sign of their participation in the pathological events observed in EVD. Of note, MBL protein, which can bind to EBOV glycoprotein, was also downregulated. It has been shown to be a neutralizing protein for EBOV, as well as being responsible for the enhancement of infection in the context of low complement [37,38]. Interestingly, low MBL levels are also associated with sepsis syndrome [39], adding a new argument in support of a sepsis-like syndrome in severe EVD cases. Finally, heat-shock proteins have also been implicated in EVD. It was thus suggested that the inhibition of HSP90 may reduce EBOV replication [40]. Here, we observed elevated levels of this protein in the acute phase of the disease, this could suggest a deleterious contribution to the outcome. HSP70 appears to be able to stimulate the immune system via TLRs and its upregulation may thus participate in inflammatory disorders [41]. Our data demonstrate global dysregulation of the biological functions closely related to innate immunity and inflammation, contributing to the loss of hemostasis.

Several treatments against EBOV have been proposed. Neutralizing antibodies (Nabs) were successfully tested in NHPs and some, to a lesser extent, in humans [42–44]. The antibody cocktail REGN-EB3 has been approved by the FDA in October 2020; it reduces mortality rate from 51% to 34%. Several antiviral molecules, such as favipiravir, have also shown some efficacy in reducing mortality in NHPs [5,45] but failed to demonstrate a resounding benefit in clinical trials [2,42]. Because of evident ethical reasons, the protocols conducted with NHPs comprise few animals and the mortality observed is to be considered with reservation because of the euthanasia of animals with acute syndrome. The results observed in studies such as the one we analyzed here is however very informative in the understanding of the parameters involved in the outcome and in the evaluation of treatment efficiency. Further studies would be needed to improve the dose regimen and the kinetic of treatment [6]. Indeed, patients who came to seek care during the epidemics probably arrived too late after symptom onset, with already a massive viral load, to allow the antiviral drugs and NAbs to significantly counteract the progression of the disease. Indeed, the drugs tested were always

much more efficient on patients with low viral loads but their efficiency dramatically dropped when administering is delayed after disease onset [42,46]. For patients in the acute phase of the disease at admission it would probably be more efficient to act on both viral load and deleterious host response [22,46–48]. The drop of lethality observed during the 2014–2016 West African EBOV outbreak in EVD patients hospitalized in intensive care units in northern countries (18%) relative to that of patients cared for in Ebola Treatment Centers in the epidemic area (63%) demonstrate the beneficial role of supportive care and thus, is consistent with the therapeutic value of host response mitigation [49,50]. In patients with a poor prognosis, EVD is accompanied by a cytokine storm evocative of septic shock syndrome [11,51,52]. Combined therapy aiming to reduce both the intensity of the dysregulated inflammatory response and the viral load would probably help patients to recover from EVD. The longer survival observed for patients treated with favipiravir during the clinical trial [53] is in accordance with our findings in NHPs. This observation indicates that antiviral treatments would probably not be sufficient to cure a highly acute infection such as EVD, at least in the epidemic setting, when initiated after disease onset. Treatment combining the administration of an antiviral, supportive care, and the mitigation of pathogenic host responses probably represents the key to curing EBOV infection.

## Supporting information

**S1 Fig. Expression of soluble CD40L.** sCD40L was measured at each sampling point during the course of the experiment by Luminex assay. The animals tested and the representation of the values are the same as in Fig 1.
(PDF)

**S2 Fig. Heatmap representing the individual values of all proteins from the inflammatory pathway.** Peptides are ranked based on the significance of the regulation of the protein, according to group and the timepoint. The color indicates the evolution of the protein level: red corresponds to upregulation and blue to downregulation.
(PDF)

**S3 Fig. Heatmap representing the individual values of the proteins involved in the coagulation pathway.** The data is presented as in S2 Fig.
(PDF)

**S4 Fig. Heatmaps representing the individual values of the proteins involved in the complement pathway and in the response to stress.** A. Heatmap representing the level of proteins from the complement pathway. B. Representation of the stress-response pathway. The representation and colors for the two heatmaps are as in S2 Fig.
(PDF)

## Acknowledgments

Experiments were carried out in the Jean Merieux–Inserm BSL4 Laboratory. We would like to thank the members of the in vivo team, S. Barron, L. Barrot, A. Duthey, F. Jacquot, M. Langry, and A. Vallve, for conducting the animal experiments. We are also grateful to A. Bocquin, S. Godard, S. Mely, E. Moissonnier, S. Mundweiler, D. Pannetier, and D. Thomas from the in vitro team, who helped to prepare the samples during the experiments and performed virus titrations. We finally thank Jean Armengaud from the Commissariat à l'Energie Atomique (CEA) for performing the mass spectrometry experiments.

## Author Contributions

**Conceptualization:** Stéphanie Reynard, Sylvain Baize.

**Formal analysis:** Stéphanie Reynard, Emilie Gloaguen, Nicolas Baillet.

**Methodology:** Jimmy Mullaert.

**Project administration:** Vincent Madelain, Hervé Raoul, Xavier de Lamballerie, Sylvain Baize.

**Supervision:** Jérémie Guedj, Sylvain Baize.

**Writing – original draft:** Stéphanie Reynard.

**Writing – review & editing:** Sylvain Baize.

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
