## [Decision Letter · Decision Letter 0]

16 Sep 2020

Dear Dr. Baize,

Thank you very much for submitting your manuscript "Early control of viral load by favipiravir promotes survival to Ebola virus challenge and prevents cytokine storm in non-human primates" for consideration at PLOS Neglected Tropical Diseases. As with all papers reviewed by the journal, your manuscript was reviewed by members of the editorial board and by several independent reviewers. In light of the reviews (below this email), we would like to invite the resubmission of a significantly-revised version that takes into account the reviewers' comments. 

We cannot make any decision about publication until we have seen the revised manuscript and your response to the reviewers' comments. Your revised manuscript is also likely to be sent to reviewers for further evaluation.

Sincerely,

Hailey Schultz

Staff Admin

Waleed Al-Salem

Deputy Editor

Reviewer's Responses to Questions

**Key Review Criteria Required for Acceptance?**

**Methods**

-Are the objectives of the study clearly articulated with a clear testable hypothesis stated?

-Is the study design appropriate to address the stated objectives?

-Is the population clearly described and appropriate for the hypothesis being tested?

-Is the sample size sufficient to ensure adequate power to address the hypothesis being tested?

-Were correct statistical analysis used to support conclusions?

-Are there concerns about ethical or regulatory requirements being met?

Reviewer #1: see comments below

Reviewer #2: (No Response)

**Results**

-Does the analysis presented match the analysis plan?

-Are the results clearly and completely presented?

-Are the figures (Tables, Images) of sufficient quality for clarity?

Reviewer #1: see comments below

Reviewer #2: (No Response)

**Conclusions**

-Are the conclusions supported by the data presented?

-Are the limitations of analysis clearly described?

-Do the authors discuss how these data can be helpful to advance our understanding of the topic under study?

-Is public health relevance addressed?

Reviewer #1: see comments below

Reviewer #2: (No Response)

**Editorial and Data Presentation Modifications?**

Reviewer #1: see comments below

Reviewer #2: (No Response)

**Summary and General Comments**

Reviewer #1: The study by Reynard et al examines three cohorts of NHPs that were infected with EBOV- an untreated control group, a group treated with favipiravir that died and a group treated with favipiravir that survived. They examined viral loads, lymphocyte and platelet counts, LFT’s and BUN, a series of cytokines and chemokines, a marker of activation on CD4 and CD8 T cells, and the proteomic blood profiles amongst these three groups. 

The real strength of this study is the ability to define factors that are associated with survival in the intervention group. Since classically EBOV infection in NHP’s is essentially a 100% lethal model, prior studies have been merely descriptive. 

Strikingly, the only thing that really distinguished the surviving treated animals from the dying treated ones was a modest (about 1 log) decline in viral load peak. No significant differences were noted in any of the other measured parameters. All of the treated animals have improved chemistries and inflammatory responses, with the exception of perforin, IL-2 and MCP-1, however, none of these markers correlated with survival. 

This is a disappointing, but important, finding: you can decrease the “inflammatory storm” but still not save the patient. This argues against the premise that disease is largely immune mediated. 

Comments:

Novel to this work is the use of proteomics in the evaluation of EBOV infection, but the methods that underly the analysis of this data were not clear. Eg line 274 a p value is assigned to day 0 in the high viremia group that indicates that “the mean parameter at day 0 was not null”. Can you please clarify? b/c it wasn’t zero it gets a p value? What is it compared to? Why is the low-viremia compared to the high viremia at time 0 but not at other time points? Why are later time points not compared low vs high but instead compared to other time points within each group? All protein abbreviations need to have definitions. 

Why was CD40L not evaluated? It has been associated with survival in humans and is easily measured in NHPs and has been done in many other Ebola NHP studies. 

Consider updating some of the refrs. The 2011 Lancet Ebola review is quite out of date given all that was learned in the 2014 outbreak. 

Figure 1: Define G/L. What does “G” mean? This is typically reported as # cells/volume

BUN is referred to as a measurement of kidney function, while this is true, elevated Bun can also reflect dehydration. 

What is the utility in showing the same data twice in figure 5C and 6A? 

It is mentioned that one animal did not have detectable viremia. Was it proven that this animal was actually infected- ie by seroconversion? Or did it have RNAemia, just not viremia? Please clarify. 

All figures were very blurry. On figure 5 there was no way to distinguish purple dots from black dots and even the red ones were tough. Can you try different shapes or some other way to make this readable? The arrows showing cytokines were also not distinguishable. It is also not clear what the direction of the arrow indicates. 

The conclusions regarding coagulation are overstated and over simplified. Line 296: “…indicating the consumption of coagulation factors”. Line 382 “Thus, most of the coagulation factors were consumed…”. The intrinsic and extrinsic coagulation pathways consist of many different factors and in your supplementary figure many of these were not significantly different in the viral load comparison groups- ie Factors 2,7,9,10,11,12 and 13- all part of the intrinsic or common pathway- all were not-significant. Factor V in the common pathway actually increased. And FGG (a subunit of factor 1) also increased, not consistent with a consumptive coagulopathy. Consider consultation with a hematologist in interpretation of this data. 

Line 376 “persistence [of cellular response] after treatment may partially explain lack of efficacy”. The fact that you found no significant differences in T cell cytokines or T cell CD69 between fatal and non-fatal treated groups argues against this conclusion. Are there other studies of T cells that you could do to assess this? Perhaps sequencing of the TCR repertoire like has been done in humans? If cellular immunity is the answer, how would favipirivir treatment modulate this physiologically? 

Line 397 “elevated levels of [HSP90] in the acute phase…and thus its upregulation does not positivity contribute to outcome”. This is correlative data only, no contribution is demonstrated. 

Line 407-410. An argument is made that it would be more efficient to use drugs that act on host response than on viral load. I would argue that the overwhelming success of mab therapeutics as noted in your ref 39 demonstrates that anti-virals are effective. Additionally, your data suggest that it might not be the inflammation that is the major factor- since the treatment group that died had a major decrease in inflammation (and other parameters you measured) but still died. 

Line 413- which aspect of “supportive care” are you referring to that demonstrates the therapeutic value of host response mitigation? Please clarify.

Reviewer #2: (No Response)

PLOS authors have the option to publish the peer review history of their article (what does this mean?). If published, this will include your full peer review and any attached files.

Reviewer #1: No

Reviewer #2: Yes: Roua Abdullah Alsubki
---

## [Decision Letter · Decision Letter 1]

1 Dec 2020

Dear Dr. Baize,

Thank you very much for submitting your manuscript "Early control of viral load by favipiravir promotes survival to Ebola virus challenge and prevents cytokine storm in non-human primates" for consideration at PLOS Neglected Tropical Diseases. As with all papers reviewed by the journal, your manuscript was reviewed by members of the editorial board and by several independent reviewers. The reviewers appreciated the attention to an important topic. Based on the reviews, we are likely to accept this manuscript for publication, providing that you modify the manuscript according to the review recommendations. 

Sincerely,

Waleed Saleh Al-Salem, Ph.D

Deputy Editor

Waleed Al-Salem

Deputy Editor

Reviewer's Responses to Questions

**Key Review Criteria Required for Acceptance?**

**Methods**

-Are the objectives of the study clearly articulated with a clear testable hypothesis stated?

-Is the study design appropriate to address the stated objectives?

-Is the population clearly described and appropriate for the hypothesis being tested?

-Is the sample size sufficient to ensure adequate power to address the hypothesis being tested?

-Were correct statistical analysis used to support conclusions?

-Are there concerns about ethical or regulatory requirements being met?

Reviewer #1: N/A

Reviewer #3: (No Response)

Reviewer #4: 1. One of the treated animals that was euthanized during the course of the protocol presented injuries. Were they unexplained hemorrhaging, bleeding or bruising? 

2. Why have you chosen to study biochemical (liver and kidney function), virological (Viral loads and viral titers), and hematological parameters (Lymphocyte and platelet counts)? Is there a direct link to the infection? Or the drug? Why not other parameters?

3. In Flow cytometry you were unable to analyze samples from day 3 and changed the protocol? Any reason for that?

**Results**

-Does the analysis presented match the analysis plan?

-Are the results clearly and completely presented?

-Are the figures (Tables, Images) of sufficient quality for clarity?

Reviewer #1: N/A

Reviewer #3: (No Response)

Reviewer #4: 1. What is the rationale behind dividing your tested animals into control (n = 10), treated and fatally-infected (n = 4), and treated & surviving (n = 5) ?

2. Pictures are blurry and difficult to see; they may consider including tables instead of some graphs.

**Conclusions**

-Are the conclusions supported by the data presented?

-Are the limitations of analysis clearly described?

-Do the authors discuss how these data can be helpful to advance our understanding of the topic under study?

-Is public health relevance addressed?

Reviewer #1: N/A

Reviewer #3: (No Response)

Reviewer #4: 1. Can you propose mechanisms of favipiravir MOA or pathogenic events observed during Ebola virus disease? In other words, is there specific biochemical interaction through which favipiravir produces its pharmacological effect that could be linked to a specific pathogenic events observed during Ebola virus disease?

2. You mentioned: favipiravir decreases the viral load and limits the flood of PAMPs, resulting in lower inflammatory responses and reduced pathology. Do you have a supporting evidence from the literature?

3. The control animals synthesized significantly higher levels of chemokines on day 7 than those that were treated, except for 

4. I was expecting to read some discussion about IL8, which was not strongly expressed in most of the controls? Why? 

5. I couldn’t see any limitations to their study nor opportunities of future work, it is recommended to conclude the discussion with such remarks.

**Editorial and Data Presentation Modifications?**

Reviewer #1: N/A

Reviewer #3: (No Response)

Reviewer #4: 1. Find another place for (Fig S4 : Expression of soluble CD40L) not in the discussion.

**Summary and General Comments**

Reviewer #1: The authors have addressed my comments/concerns.

Reviewer #3: This manuscript describes the evaluation of multiple virological, immunological and biochemical parameters in non-human primates, either untreated or treated with favipiravir, following Ebola virus infection. The manuscript is well-written, but I do think the authors could better put their current findings into context with previously published work. Also, as with any observational study like this one, the authors should take care not to over-state their conclusions. I have some suggestions to change some of this language listed in the minor points below.

Major points:

There have been multiple studies looking at EBOV-infection in non-human primates. I think it would be good for the authors to add some more language to put their findings into context with that already in the literature: what findings reported here are new, which agree or disagree with previous reports etc.

In particular, please compare with the results described in the authors’ previous paper Madelain et al (2018; PMID 30275474), where some of the same parameters were measured. Is this study with different samples or were they processed in a different way? Fig.2 from that paper shows IL6, IFNα, TNFα depicted on a log scale, whereas the current study shows them on a linear scale, so it is difficult to directly compare. The current study does stratify the animals by survival rather than treatment regimen, but if the data is the same as previously published, then this should be stated.

Minor points:

Line 73: To save the reader having to go back and look up these references, here it would be convenient to state the favipiravir treatment regimen the animals received, ie what dose and route, how often, and in what relation to the time of infection.

Line 229: “Cytokine expression is directly linked to viremia”, please change to “Cytokine expression is correlated with viremia”.

Line 241: “Cytokine levels are therefore dependent on viral replication”, please change to “Cytokine levels are therefore correlated with viral replication.”

Line 363: “These results demonstrate that reduction of the viral load by favipiravir treatment resulted in decreased cytokine/chemokine release.”, suggest changing to “These results demonstrate that reduction of the viral load by favipiravir treatment correlated with decreased cytokine/chemokine release.”

Line 372: “By acting on viral replication, favipiravir decreases the viral load and limits the flood of PAMPs…”, suggest changing to “We hypothesize that by acting on viral replication, favipiravir decreases the viral load and limits the flood of PAMPs…”

Line 375: Unless I’m misunderstanding what is meant here, the sentence starting “Similarly…” seems to actually be opposite of the previous sentence? Please clarify.

Line 380: Please change “directly results from” to “correlates with”.

Line 380: “…this improvement”, I think should read “improving these”.

Line 388: “…the lack of efficacy…”, change to “the incomplete efficacy…”.

Line 420: A mAb-based treatment for EBOV has recently been approved by FDA. Please update to reflect this.

Line 437: I would take issue with this statement, as the recently FDA-approved mAb therapy did indeed show clinical benefit “in the epidemic setting, when initiated after disease onset”, albeit with incomplete efficacy. Please soften this statement.

Reviewer #4: I am wondering is this study a part of a clinical trials? Or is there a plan to conduct one? All in all, the study is excellent and their findings are helpful to advance our understanding of the topic under study

PLOS authors have the option to publish the peer review history of their article (what does this mean?). If published, this will include your full peer review and any attached files.

Reviewer #1: No

Reviewer #3: No

Reviewer #4: Yes: Bandar Alosaimi
---

## [Decision Letter · Decision Letter 2]

9 Mar 2021

Dear Dr. Baize,

We are pleased to inform you that your manuscript 'Early control of viral load by favipiravir promotes survival to Ebola virus challenge and prevents cytokine storm in non-human primates' has been provisionally accepted for publication in PLOS Neglected Tropical Diseases.

Best regards,

Waleed Saleh Al-Salem, Ph.D

Deputy Editor

Waleed Al-Salem

Deputy Editor

Reviewer's Responses to Questions

**Key Review Criteria Required for Acceptance?**

**Methods**

-Are the objectives of the study clearly articulated with a clear testable hypothesis stated?

-Is the study design appropriate to address the stated objectives?

-Is the population clearly described and appropriate for the hypothesis being tested?

-Is the sample size sufficient to ensure adequate power to address the hypothesis being tested?

-Were correct statistical analysis used to support conclusions?

-Are there concerns about ethical or regulatory requirements being met?

Reviewer #1: Comments have been adequately addressed.

Reviewer #4: (No Response)

**Results**

-Does the analysis presented match the analysis plan?

-Are the results clearly and completely presented?

-Are the figures (Tables, Images) of sufficient quality for clarity?

Reviewer #1: Comments have been adequately addressed.

Reviewer #4: (No Response)

**Conclusions**

-Are the conclusions supported by the data presented?

-Are the limitations of analysis clearly described?

-Do the authors discuss how these data can be helpful to advance our understanding of the topic under study?

-Is public health relevance addressed?

Reviewer #1: Comments have been adequately addressed.

Reviewer #4: (No Response)

**Editorial and Data Presentation Modifications?**

Reviewer #1: (No Response)

Reviewer #4: (No Response)

**Summary and General Comments**

Reviewer #1: Comments have been adequately addressed.

Reviewer #4: (No Response)

PLOS authors have the option to publish the peer review history of their article (what does this mean?). If published, this will include your full peer review and any attached files.

Reviewer #1: No

Reviewer #4: **Yes: **Dr Bandar Alosaimi

---

## [Editor Report · Acceptance letter]

25 Mar 2021

Dear Dr. Baize,

We are delighted to inform you that your manuscript, "Early control of viral load by favipiravir promotes survival to Ebola virus challenge and prevents cytokine storm in non-human primates," has been formally accepted for publication in PLOS Neglected Tropical Diseases.

Best regards,

Shaden Kamhawi

co-Editor-in-Chief

Paul Brindley

co-Editor-in-Chief
